# Descriptions of Two New Species, *Sillago muktijoddhai* sp. nov. and *Sillago mengjialensis* sp. nov. (Perciformes: Sillaginidae) from the Bay of Bengal, Bangladesh

**Shilpi Saha** [1,2]**, Na Song** [1]**, Zhengsen Yu** [1]**, Mohammad Abdul Baki** [2]**, Roland J. McKay** [3]**, Jianguang Qin** [4] **and Tianxiang Gao** [5,*]

1    Fisheries College, Ocean University of China, Qingdao 266003, China; shilpisahazoo@gmail.com (S.S.); songna624@163.com (N.S.); yuzhengsen01@126.com (Z.Y.)
2    Department of Zoology, Jagannath University, Dhaka 1100, Bangladesh; mabaki@gmail.com
3    Chillagoe Museum, Mareeba 4101, Australia; rolandmckay@yahoo.com.au
4    School of Biological Sciences, Flinders University, Adelaide 5001, Australia; jian.qin@flinders.edu.au
5    Fishery College, Zhejiang Ocean University, Zhoushan 316022, China
*    Correspondence: gaotianxiang0611@163.com

**Abstract:** Due to difficulty in recognition, many true species have been covered under the synonyms of wide-spread species. To justify the identification of a widely distributed species, *Sillago sihama* from the Bay of Bengal, Bangladesh, an integrated approach including morphology and DNA barcoding was used. Two unrecognized species of *Sillago*, i.e., *Sillago muktijoddhai* sp. nov. and *S. mengjialensis* sp. nov., were identified from the coastal area of Bangladesh. *S. muktijoddhai* sp. nov. has marked differences in the body color, anal fin color, number of gill rakers, snout length, and swimbladder. *S. mengjialensis* sp. nov. has notable differences in the anal fin color, snout length, and swimbladder and is distinguished from *S. muktijoddhai* sp. nov. by the body color and swimbladder. The morphological characters of 14 documented *Sillago* species with two posterior extensions of the swimbladder were referenced and distinguished to accredit the two new species. Genetic analyses of partial mitochondrial cytochrome oxidase subunit I and 16S ribosomal RNA also supported the validity of the new species. This study has increased the number of recognized species of *Sillago* in the world and confirmed the prevailing misidentification of these two new species in Bangladesh as so-called *S. sihama*. Moreover, the study confirmed the misidentification of *S. mengjialensis* sp. nov. in Indonesia as *S. sihama* and the identification of unknown *Sillago* sp.1 in India.

**Keywords:** Bay of Bengal; DNA barcoding; morphology; new species; Sillaginidae; *Sillago*

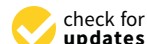

## 1. Introduction

Fishes of the family Sillaginidae Richardson, 1846 generally inhabit inshore coastal waters or estuarine areas of rivers with open sandflats and muddy substrates. They commonly feed on crustaceans and molluscs, and various larger fish, marine mammals, and seabirds are their predators. The species of this family are widely distributed in the Indo-West Pacific region [1–3]. Presently, the family includes 39 species and 5 genera [3–13]. Nine species of *Sillago*, i.e., *S. sihama,* and *S. chondropus* [14], *S. vincenti* [15], *S. intermedius* [16], *S. indica* [17], *S. lutea, S. ingenuua* [1], *S. soringa,* and *S. aeolus* [18,19], and one species of *Sillaginopsis*, i.e., *S. panijus,* are distributed in the Bay of Bengal [14]. Thus far, investigations have been exclusively based on morphological characters. Furthermore, very few studies have focused on the taxonomy of Sillaginidae from the Bay of Bengal, Bangladesh. The short descriptions of *Sillaginopsis panijus*, *S. maculata*, and *S. sihama* were based on external morphological characters only without any studies of the swimbladder structure, vertebrae, or molecular markers [20,21]. *S. sihama* is a cryptic species complex, and *S. maculata* is an indigenous species in Australia at present, distributed only along the east coast of Australia [22,23]. These two species may be incorrectly identified in Bangladesh.

The structural differences in the swimbladder and the division of the vertebrae at the abdominal, modified (haemal), and caudal positions were important for the taxonomic identification of some cryptic *Sillago* species [1,5]. However, McKay (1985) proposed swimbladder features for the subgenera-grading system of the genus *Sillago* including three subgenera: *Sillaginopodys* Fowler, 1933 (reduced swimbladder, no duct-like process); *Sillago* Cuvier, 1817 (two tapering extensions arise from the posterior portion of the swimbladder, duct-like process present); and *Parasillago* (a single extension arises posteriorly from swimbladder and the duct-like process present) [1]. Even so, the very similar swimbladders of some sibling species (e.g., *S. shaoi* and *S. sihama*) have made their identification difficult. Using only phenotypic data, six newly diagnosed *Sillago* species were incorrectly identified as *S. sihama* [6–11]. Recently, in the Indo-West Pacific region, by using phenotypic traits and molecular markers, eight cryptic lineages were recognized in the *S. sihama* complex [22].

Therefore, the integrated approach including studies of key morphological characters of sillaginids and DNA barcoding will be helpful for the identification and confirmation of cryptic sillaginids species [5,24]. A mitochondrial protein-coding gene, cytochrome oxidase subunit I (COI), was used for DNA barcoding and to investigate the phylogenetic relationships of the sillaginids species [7,9]. In addition, together with COI, 16S ribosomal RNA (16S rRNA) can be used for the identification of species that share morphological similarities [22].

The present study aims to describe two new *Sillago* species from the Bay of Bengal, Bangladesh, using a comparison of morphometric and meristic data, coloration patterns, number of vertebrae, swimbladder morphology, and molecular markers. The results will be beneficial for the identification of Sillaginidae species that are important for fishery management, biodiversity maintenance, and resource exploitation in the Bay of Bengal, Bangladesh.

## 2. Materials and Methods

A total of 211 fresh specimens of 2 new *Sillago* species were directly collected from local fishermen who caught these fishes using beach seines from the Cox's Bazar, Sundarbans, Patharghata, Maheshkhali and Saint Martin's Island (Figure 1). *S. muktijoddhai* sp. nov. was collected from all five sites, and *S. mengjialensis* sp. nov. was collected from three sites during October 2018 to February 2020; the detailed sampling information can be found in Supplementary Table S1. Type specimens were preserved in 95% ethanol under −20 °C. Preserved specimens were specified with the respective holotype and paratypes.

The genus and species classification and terminology followed the work of McKay (1992) and Kaga (2013) [5,25]. The terms used to describe appendages of the swimbladder followed those of other researchers [22,26,27]. The definition of the modified vertebrae followed McKay (1992) [5]. Except for the swimbladder and vertebrae, 11 meristics and 19 morphometrics characters were considered, and some descriptive characters such as body and fin coloration were also used. The number of vertebrae was counted by radiograph (3–6 individuals, Supplementary Figure S1) and dissection (the remaining individuals) of each species.

COI and 16S rRNA fragments were amplified to analyze genetic differences between sillaginids. Epaxial white muscle tissue was collected from 26 fresh specimens of 4 Sillaginidae species and preserved in 95% ethanol under −20 °C. Genomic DNA was extracted by proteinase K digestion followed by the standard phenol–chloroform method [28]. The primers used were as follows: COI Primer—FishF1: 5′-TCAACCAACCACAAAGACAT TGGCAC-3′ and FishR1: 5′-TAGACTTCTGGGTGGCCAAAGAATCA-3′, and 16S rRNA Primer—16S-arF: 5′-CGCCTGTTTATCAAAAACAT-3′ and 16S-brR: 5′-CCGGTCTGAACTC AGATCACGT-3′ [29]. The PCR reaction system was 25 µL containing 1 µL template DNA, 2.5 µL of 10× PCR buffer, 1.5 mmol/L $MgCl_2$, 200 µmol/L dNTPs, 0.2 mmol/L of each primer, and 1.25 units of Taq DNA polymerase; the reaction conditions were predenaturation at 94 °C for 5 min; denaturation at 94 °C for 45 s; annealing at 50 °C for 45 s; extension at 72 °C for 45 s; 35 cycles in total; and a final extension at 72 °C for 10 min.

The PCR products were separated on 1.5% agarose gel and cleaned with the BioDev Gel Extraction System B (BioDev Technology, Beijing, China). The cleaned products were sequenced by a BigDye Terminator cycle sequencing kit v2.0 (Applied Biosystems, Foster City, CA, USA), and sequencing was performed on an ABI Prism 3730 automatic sequencer (Applied Biosystems) with both forward and reverse primers used for amplification.

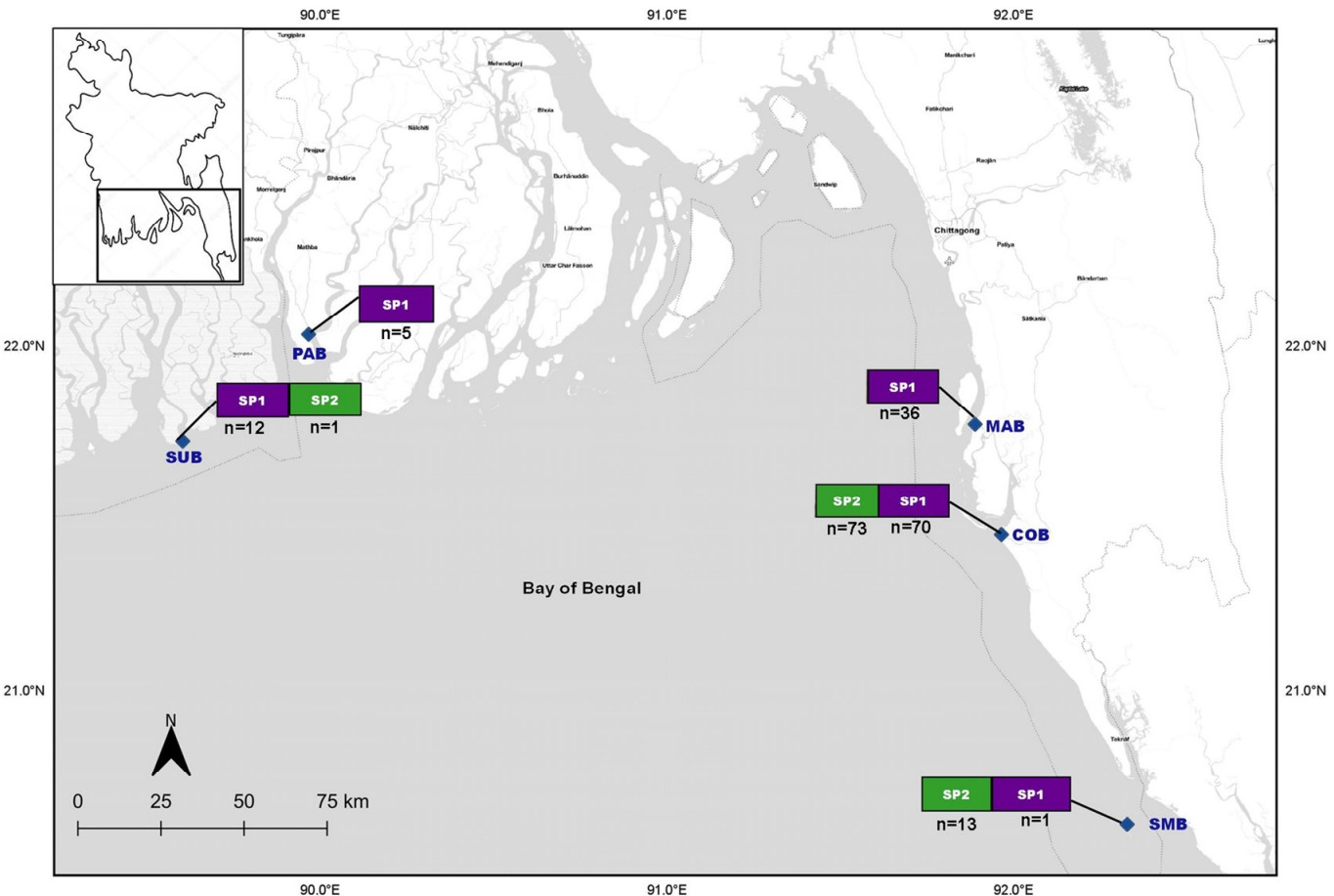

**Figure 1.** Sampling locations of two *Sillago* spp. nov. from the Bay of Bengal, Bangladesh. SUB: Sundarbans, PAB: Patharghata, MAB: Maheshkhali, COB: Cox's Bazar, and SMB: Saint Martin's Island, SP1: S. *muktijoddhai* sp. nov., SP2: S. *mengjialensis* sp. nov.; n is the number of individuals of each species at each site.

Lasergene software (Lasergene, Madison, WI, USA) was used to compare forward and reverse sequences. Clustal X 2.1 was used to align the consensus sequences [30]. The best evolution model of nucleotide substitution (i.e., the GTR+I+G model) was selected by jModelTest v2.1.10 [31]. The maximum likelihood and Bayesian inference methods were used to infer phylogeny considering their advantages. The maximum likelihood method evaluates different tree topologies, uses all the sequence information, and is least affected by sampling error; posterior probability in the Bayesian inference method is associated with its probability of being correct, given the prior probability, model, and data. MEGA 6.0 was used to perform maximum likelihood analysis with 1000 bootstrap replications [32]. The Bayesian phylogenetic analysis was carried out using MrBayes v.3.2.6 with default parameters except where otherwise mentioned [33]. Analyses were run with Markov Chain Monte Carlo (MCMC) sampling with four chains. Trees and parameters were sampled every 100 generations over ten million generations, with the first 25% of the samples discarded as burn-ins. A tree with Bayesian posterior probabilities was visualized in FigTree v1.4.4.

Only haplotype sequences were deposited in GenBank. Almost all the available sillaginids sequences (COI:107, 16S rRNA: 36) downloaded from GenBank are cited in the present study. The accession numbers of the sequences are mentioned in the relevant figures, and detailed information can be found in Supplementary Table S2 [7,9,11,12,22,24,34–44].

## 3. Results

### *3.1. Sillago muktijoddhai* Gao *and* Saha sp. nov.

#### 3.1.1. Holotype

FELOUC142377; 92 mm standard length (SL); Cox's Bazar, Bangladesh; collected by Shilpi Saha, February 2019; deposited at Fishery Ecology Laboratory, Fisheries College, Ocean University of China (FELOUC), Qingdao, China.

#### 3.1.2. Paratypes

FELOUC142372-75, CO219-6, 21, 67, 70,71, 73–78, 80, 82, 84, 90–93, 95, 96, 99–101, 104, 106–109, 111–113, and 115; 36 individuals; 82.03–123.12 mm SL; collection data and deposition same with holotype; FELOUC142382-85, 87–92; 10 individuals; 77–94 mm SL; Sundarban (Dublarchar), Bangladesh; collected by Mohammad Abdul Baki, February 2019; deposited at FELOUC; CO12052-61; 10 individuals; 60–74 mm SL; Cox's Bazar, Bangladesh; collected by Kishor Kumar Sarker, November 2019; deposited at Fisheries Laboratory, Department of Zoology, Jagannath University (FLJNU), Dhaka, Bangladesh.

#### 3.1.3. Etymology

The word '*muktijoddhai*' is derived from the Bengali word '*muktijoddha*' meaning a freedom fighter of Bangladesh in 1971. The specific name '*muktijoddhai*' was chosen in honor of them.

#### 3.1.4. Diagnosis

*S. muktijoddhai* sp. nov. is distinguished by X–XI, I+20–22 dorsal-fin rays; II+21–23 anal-fin rays; 68–72 scales in the lateral line; 4–6 scales above the lateral line; 3–5/8–10 gill rakers on the first arch; vertebra: abdominal 12–14 (mostly 13), modified 5–8 (mostly 8), caudal 12–15 (mostly 13), and total 32–35 (mostly 34). The body is greenish dorsally and light yellowish ventrally; there is a presence of black dots on the anal fins and two or three rows of dark spots on the second dorsal-fin membrane (Figure 2a); the swimbladder is short and broad with two anterior extensions and two posterior extensions with lacuna at the base; the anterolateral extension extends into an anterior short, blind tubule, and the posterior one is kinked, long, and complex for about half of its length towards the beginning; there are 8–9 lateral processes.

#### 3.1.5. Description

Counts and measurements are given in Table 1. The body is elongated, slightly steep anteriorly, and tubular posteriorly. The upper jaw is slightly protracted and crescentic, with minute villiform teeth on both jaws in one row. The back edge of the preopercle is slightly denticulated, and the opercle has one fragile spine posterodorsally. Gill rakers on the first arch are pointed and gradually become short towards the end. The body is covered with moderate-sized, overlapping ctenoid scales. The cheek scales cycloid, arranged in two rows.

There are two distinctly separate dorsal fins. The first dorsal fin is higher than the second, originating above the pectoral-fin base; its second spine is the longest, and the length of the succeeding spines decrease gradually. The base of the second dorsal fin is long, beginning at the midbody and not reaching the caudal-fin base when depressed. The anal fin originates slightly posterior to the anus, not reaching the caudal-fin base when depressed. The two disconnected pelvic fins are wide, roughly three-cornered, and shorter than the pectoral fin.

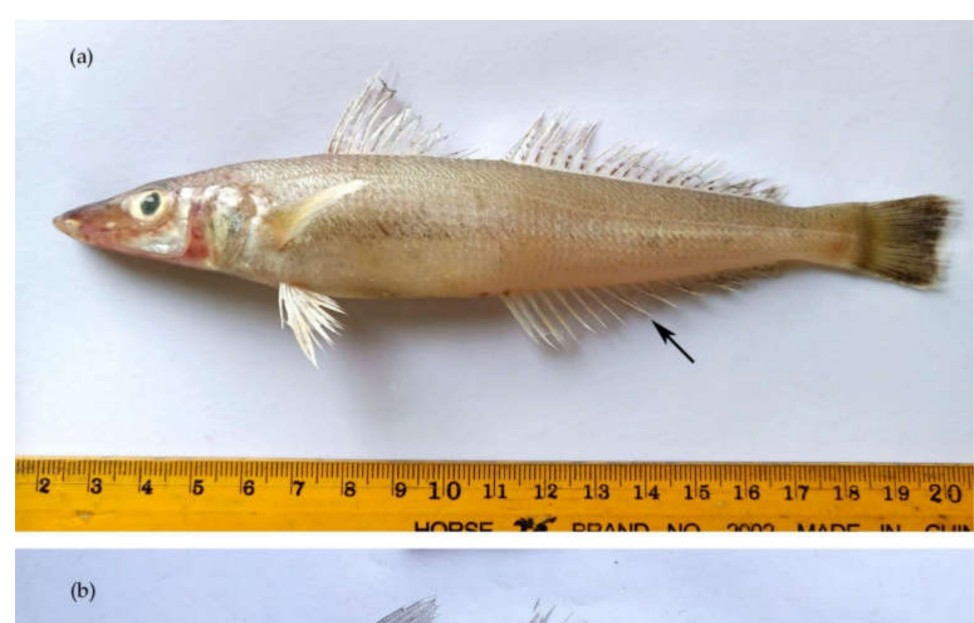

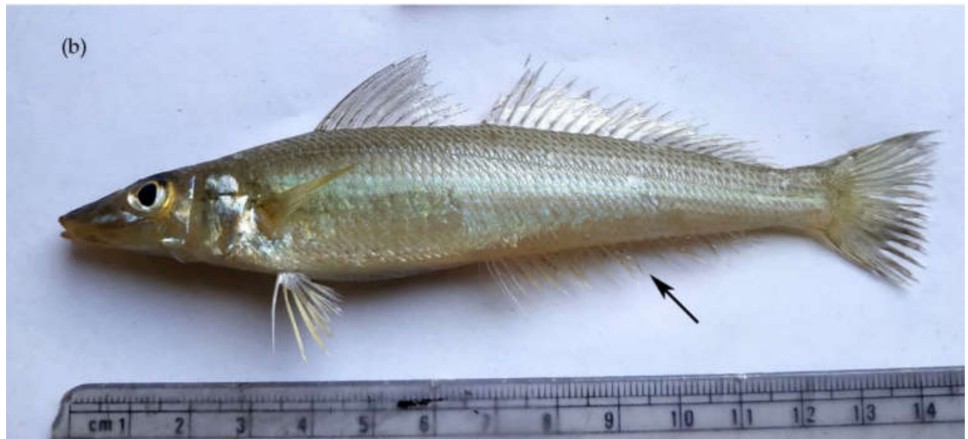

**Figure 2.** Two *Sillago* spp. nov. from the Bay of Bengal, Bangladesh. (**a**) *S. muktijoddhai* sp. nov., DS1201, paratype, 157.84 mm standard length (SL); (**b**) *S. mengjialensis* sp. nov., CO219110, paratype, 125 mm SL.

**Table 1.** Morphomeristic characters of two *Sillago* spp. nov. from the Bay of Bengal, Bangladesh.

| Meristics and Morphometric Measurements (mm) | *S. muktijoddhai* sp. nov. | | *S. mengjialensis* sp. nov. | |
|---|---|---|---|---|
| | Holotype (FELOUC142377) | Paratypes (*n* = 123) | Holotype (FELOUC142378) | Paratypes (*n* = 86) |
| Dorsal fins | XI, I+21 | X–XI, I+20–22 | XI, I+20 | XI, I+20–22 |
| Anal fin | II+22 | II+21–23 | II+22 | II+20–23 |
| Pectoral fin rays | 16 | 15–18 | 15 | 15–17 |
| Pelvic fin rays | I+5 | I+5 | I+5 | I+5 |
| Caudal fin rays | 17 | 16–17 | 17 | 16–17 |
| Scales in the lateral line | 69 | 68–72 | 68 | 66–72 |
| Scales above/below the lateral line | 5/12 | 4–6/10–13 | 5/11 | 4–5/10–12 |
| Gill rakers on first arch | 3 + 9 = 12 | 3–5 + 8–10= 11–14 | 3 + 9 = 12 | 3–4 + 8–10 = 11–13 |
| Vertebrae (AV+HV+CV) | – | 12–14 + 5–8 + 12–15 = 32–35 | – | 13–14 + 5–8 + 12–15 = 31–35 |
| Total weight (TW, g) | 6.9 | 2.28–41.59 | 8.2 | 2.1–16.9 |
| Total length (TL) | 107 | 61.36–191 | 113 | 63.1–142.54 |
| Standard length (SL) | 92 | 53.29–161 | 98 | 53.97–121.69 |
| Head length (HL) | 26 | 14.79–49 | 28 | 14.85–33.46 |
| Upper jaw length (UJL) | 5 | 3.24–8.39 | 5 | 2.50–7.73 |
| Lower jaw length (LJL) | 4 | 2.6–7.43 | 4 | 2.3–6.19 |
| Postorbital length (PL) | 10 | 4.95–21 | 10 | 5.71–13 |

**Table 1.** *Cont.*

| Meristics and Morphometric Measurements (mm) | *S. muktijoddhai* sp. nov. | | *S. mengjialensis* sp. nov. | |
|---|---|---|---|---|
| | Holotype (FELOUC142377) | Paratypes (*n* = 123) | Holotype (FELOUC142378) | Paratypes (*n* = 86) |
| Snout length (slw) | 8 | 3.3–16 | 8 | 2.91–11.0 |
| Eye diameter (ED) | 5 | 3.69–9 | 6 | 3.35–9.17 |
| Interorbital width (IW) | 4 | 2.94–8 | 5 | 2.08–7.0 |
| Caudal peduncle depth (CPD) | 6.31 | 4.11–11 | 7.31 | 3.41–9.08 |
| Caudle peduncle length (CPL) | 8 | 4.05–17.77 | 10.73 | 3.28–12.84 |
| First dorsal fin base (D1L) | 18 | 9.83–34 | 18 | 10.16–25.93 |
| Second dorsal fin base (D2L) | 33 | 20.35–59 | 35 | 20.45–44.09 |
| Anal fin base (AL) | 34 | 21.68–59 | 36 | 19.37–44.12 |
| Pectoral fin length (ptl) | 15 | 7.71–26 | 15 | 7.9–18.04 |
| Pelvic fin length (pvl) | 14 | 7.18–25 | 14 | 6.85–19.46 |
| Body width (BW) | 10 | 5.13–21 | 10 | 6.23–14.88 |
| Body depth (BD) | 16 | 8.31–26 | 17 | 8.58–19.6 |
| As % of SL | | | | |
| Body depth (BD) | 17.39 | 12.08–19.49 | 17.35 | 10.72–20.34 |
| Head length (HL) | 28.26 | 21.81–31.31 | 28.57 | 25.02–31.68 |
| Caudal peduncle length (CPL) | 8.70 | 6.66–13.13 | 10.95 | 5.11–13.25 |
| As % of HL | | | | |
| Eye diameter (ED) | 19.23 | 16.28–29.54 | 21.43 | 16.67–29.87 |
| Interorbital width (IW) | 15.38 | 13.70–21.87 | 17.86 | 13.65–25.33 |
| Snout length (SLw) | 30.77 | 20.53–35.98 | 28.57 | 17.3–37.53 |
| Postorbital length (PL) | 38.46 | 32.14–46.51 | 35.71 | 29.62–44.59 |
| DCP/LCP | 78.88 | 56.05–94.1 | 68.13 | 54.52–94.97 |

3.1.6. Color of Fresh Specimens

The body is greenish dorsally and light yellowish ventrally with black dots on the side below the lateral line. The cheek has black dots gathered on the anteroventral part of the eyes. The dorsal fins are hyaline, and small dark spots exist on the fin membrane, but those on the second dorsal fin form two or three distinct rows. The pectoral and pelvic fins are light yellowish. The anal fin is light yellowish with black spots. The caudal fin is light yellowish, dusky, and with a white edge; the lobes are truncated or emarginated.

3.1.7. Swimbladder

The swimbladder is broad (Figure 3a,c). Two anterior extensions split to the end anteriorly on each side of the basioccipital over the auditory capsule. An anterolateral extension originates anteriorly on both sides of the swimbladder and is bifurcated into anterior and posterior sub-extensions: the anterior sub-extensions are short, simple blind tubules; the posterior one is kinked, long, and complex for about half of its length towards the beginning, then the remainder is simple and thin, extending along the abdominal wall and terminating well beyond the roots of the two posterior extensions, relatively. The entire lateral surface of the main body of the swimbladder has eight or nine lateral processes that penetrate into the musculature; the anterior two to three are robust and horn-like; the posterior six or seven are rather short and triangular. The posterior sub-extensions of the swimbladder are ventrally adjacent to the lateral processes but not interconnected with them. Two posterior tapering extensions of the swimbladder extend into the caudal region. The roots of the two posterior extensions are non-adherent, and the two posterior extensions are non-adjoining with a lacuna between them. From the ventral surface of the swimbladder, a single duct-like process arises and arrives at the urogenital aperture, the

duct-like process begins at the termination of the swimbladder and is between the base of the two posterior extensions. A sub-extension is attached to a sanguineous vesicle near the vertebra.

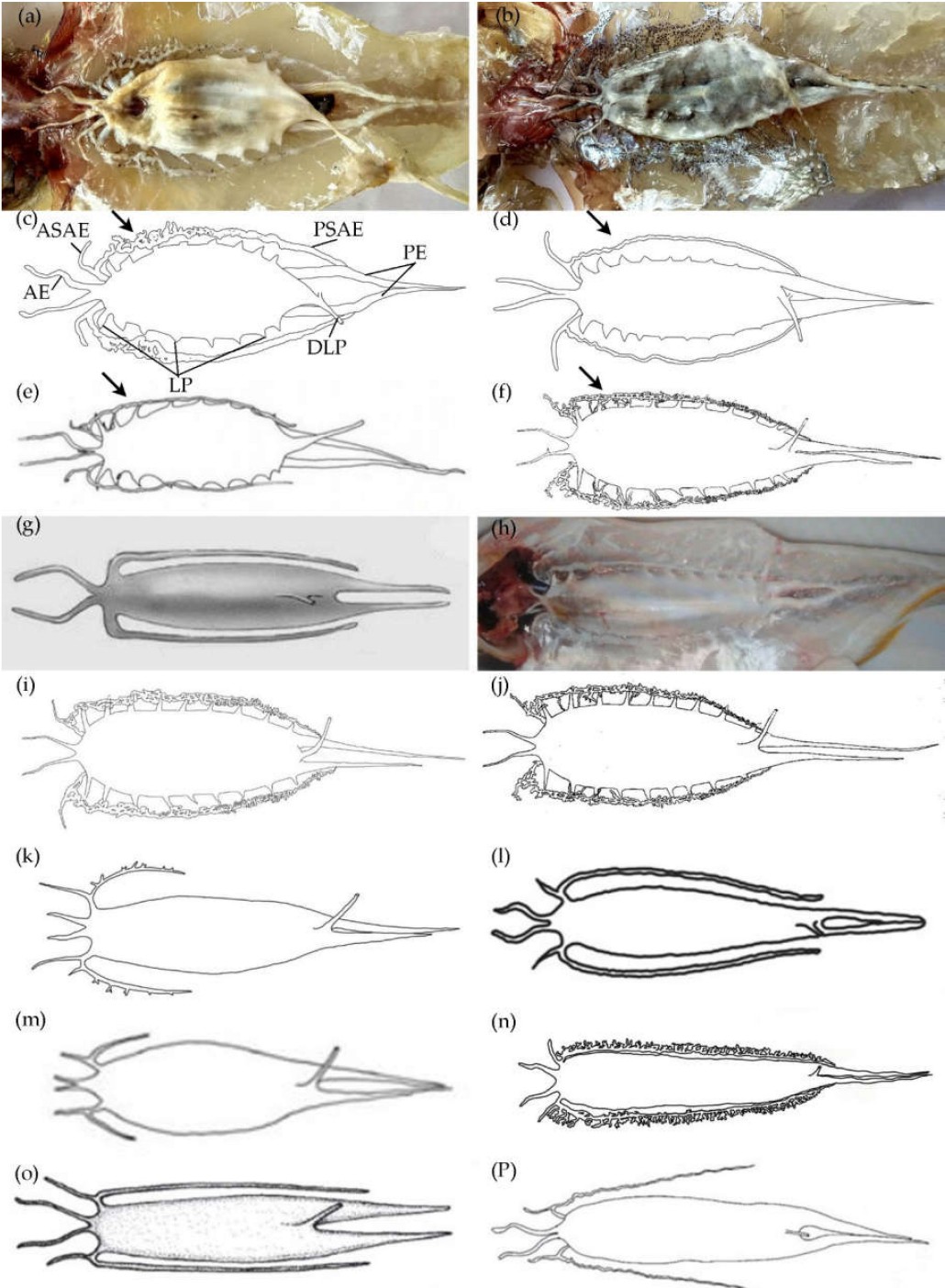

**Figure 3.** Swimbladder of two *Sillago* spp. nov. and twelve other *Sillago* species. (**a**,**c**) *S. muktijoddhai* sp. nov.; (**b**,**d**) *S. mengjialensis* sp. nov.; (**e**) *S. indica* [45]; (**f**) *S.* cf. *sihama*③(cryptic species in the *Sillago sihama* complex), China [11]; (**g**) *S. sihama*, Red Sea [8]; (**h**) *S. malabarica* [12]; (**i**) *S. shaoi* [9]; (**j**) *S. nigrofasciata* [11]; (**k**) *S. sinica* [7]; (**l**) *S. suezensis* [8]; (**m**) *S. panhwari* [22]; (**n**) *S. parvisquamis* [11]; (**o**) *S. intermedius* [5]; (**p**) *S. caudicula* [6]. AE: anterior extension, ASAE: anterior sub-extension of anterolateral extension, PSAE: posterior sub-extension of anterolateral extension, LP: lateral processes, DLP: duct-like process, PE: posterior extension. Black arrows indicate differences between new and other *Sillago* species.

### 3.1.8. Habitat

Frequently found in brackish and marine water alongside beaches and mangrove creeks with a sandy substrate and captured by nets (called dati jal or poka jal in the local language) set at the bottom of the nearshore areas.

### 3.1.9. Distribution

*S. muktijoddhai* sp. nov. have, to date, only been found along the inshore regions of the Bay of Bengal, Bangladesh, including Sundarbans (Dublarchar), Patharghata (Balashwar river), Maheshkhali (Kohelia River), Cox's Bazar, and Saint Martin's Island (Figure 1).

### 3.2. Sillago mengjialensis Gao, Baki *and* Saha sp. nov.

### 3.2.1. Holotype

FELOUC142378; 98 mm SL; Cox's Bazar, Bangladesh; collected by Shilpi Saha, February 2019; deposited at FELOUC.

### 3.2.2. Paratypes

FELOUC1423-76,79-81, 86, CO219-18, 49, 55, 68, 69, 72, 81, 83, 85–89, 94, 97, 98, 102, 103, 105, 114; 26 individuals; 78–121.69 mm SL; collection data and deposition the same as the holotype. CO12024, 29, 68–76; 11 individuals; 61.5–111 mm SL; Cox's Bazar, Bangladesh; collected by Kishor Kumar Sarker, November 2019; deposited at FLJNU.

### 3.2.3. Etymology

The specific name '*mengjialensis*' is derived from the Chinese word 'Mèngjiālā' meaning Bengali and refers to the contributions of China and Bangladesh in identifying the species.

### 3.2.4. Diagnosis

*S. mengjialensis* sp. nov. is distinguished by XI, I+20–22 dorsal-fin rays; II+20–23 anal-fin rays; 66–72 scales in the lateral line; 4–5 scales above the lateral line; 3–4 + 8–10 gill rakers on the first arch; vertebra: abdominal 13–14 (mostly 13), modified 5–8 (mostly 5), caudal 12–15 (mostly 14), and total 31–35 (mostly 34). The body is light olive green dorsally and silver ventrally (Figure 2b). There are black dots on the anal fin, and the swimbladder is long with two anterior extensions and two posterior extensions without lacunae at the base. The anterolateral extension extends into the anterior short, blind tubule, and the posterior one is kinked, long, and thin with 9–10 lateral processes.

### 3.2.5. Description

Counts and measurements are given in Table 1. The body is elongated, somewhat conical anteriorly, and cylindric posteriorly. The back edge of the preopercle is slightly toothed. The opercle only has one weak spine posterodorsally. The gill rakers on the first arch are pointed and gradually become small towards the end. The body is covered with moderate-sized, overlapping ctenoid scales. The cheek scales are cycloid, arranged in two rows.

There are two disconnected dorsal fins. The first dorsal fin is higher than the second, originating above the pectoral fin base; its second spine is the longest, and the length of the succeeding spines decreases gradually. The base of the second dorsal fin is long, beginning at the midbody, and not reaching the caudal-fin base when depressed. The anal fin originates slightly posterior to the anus, not reaching the caudal-fin base when depressed. Two disconnected pelvic fins are large, roughly trigonal in shape, and smaller than the pectoral fin.

### 3.2.6. Color of Fresh Specimens

The body is light olive greenish dorsally and silver ventrally with black spots on the sides below the lateral line. The cheek has black spots gathered on the anteroventral part of the eyes. The dorsal fins are hyaline with small dark spots on the fin membrane adjacent to

the ray. The pectoral and pelvic fins are light yellowish. The anal fin is light yellowish to whitish with black dots. The caudal fin light yellowish, dusky, and has a white margin; the lobes are truncated or emarginated.

### 3.2.7. Swimbladder

The swimbladder is long (Figure 3b,d). Two anterior extensions split to the end anteriorly on each side of the basioccipital over the auditory capsule. An anterolateral extension originates anteriorly on both sides of the swimbladder and is bifurcated into anterior and posterior sub-extensions: the anterior sub-extensions are short, simple blind tubules; the posterior one is kinky, long, and thin, extending along the abdominal wall and terminating at the roots of the two posterior extensions, respectively. The entire lateral surface of the main body of the swimbladder has nine or ten lateral processes penetrating the musculature; the anterior three or four are robust and horn-like; the posterior five or six are relatively short and triangular. The posterior sub-extensions of the swimbladder are ventrally adjacent to the lateral processes but not interconnected with them. There are two posterior tapering extensions of the swimbladder extending into the caudal region. The origins of the two posterior extensions are adherent, and the two posterior extensions are well-knit without a lacuna between them. From the ventral surface of the swimbladder, a single duct-like process arises, arriving at the urogenital aperture. The duct-like process begins at the termination of the swimbladder, between the base of the two posterior extensions. A sub-extension is attached to a sanguineous vesicle near the vertebra.

### 3.2.8. Habitat

Estuarine and marine water are mostly preferred. Commonly found in the sandy bottom of inshore areas.

### 3.2.9. Distribution

*S. mengjialensis* sp. nov. has, to date, only been found to be distributed in the coastal waters of Bangladesh, including Sundarbans (Dublarchar), Cox's Bazar, and Saint Martin's Island (Figure 1).

### *3.3. Molecular Analysis of the COI and 16S rRNA Gene*

Eleven haplotype COI and seven haplotype 16S rRNA sequences were obtained from two new *Sillago* species, *Sillaginopsis panijus* and *Sillago macrolepis*.

After sequence alignment, 544 bp and 580 bp fragments were obtained for the COI and 16S rRNA genes, respectively. The mean genetic intraspecific divergences of COI were 0.5%, and of 16S rRNA they were 0.2–0.3% in two *S.* spp. nov. The mean genetic divergences of COI for the interspecific level ranged 17.5–25.9% and 5.3–24.1% between *S. muktijoddhai* sp. nov. and 41 other sillaginids and *S. mengjialensis* sp. nov. and 41 other sillaginids, respectively (details in Supplementary Table S3). The mean genetic divergences of 16S rRNA for the interspecific level ranged 8.1–22.1% and 1.3–20.5% between *S. muktijoddhai* sp. nov. and 31 other sillaginids and *S. mengjialensis* sp. nov. and 31 other sillaginids, respectively (details in Supplementary Table S4).

Based on the COI and 16S rRNA sequences, the phylogenetic tree (Figure 4a,b and Supplementary Figure S2) showed that two newly discovered *Sillago* species individuals formed a separate clade with a previously discovered species. Based on the COI gene sequences, *S. muktijoddhai* sp. nov. is situated in the clade of misidentified *S. sihama,* revealing an intraspecies relationship with MH429345 (genetic distance of 0.5%). *S. mengjialensis* sp. nov. also clustered and confirmed the identification of the misidentified *S. sihama* from Indonesia (JN312946, divergence of 0.5%). Based on the 16S rRNA gene sequences, *S. mengjialensis* sp. nov. was first clustered with the unidentified species *Sillago* sp.1, India (KC835208, divergence 0.3%), confirming the intraspecies relationship. Then, it clustered with *Sillago nigrofasciata*; the distance was 1.3% (COI: 5.3%), confirming the interspecies relationship.

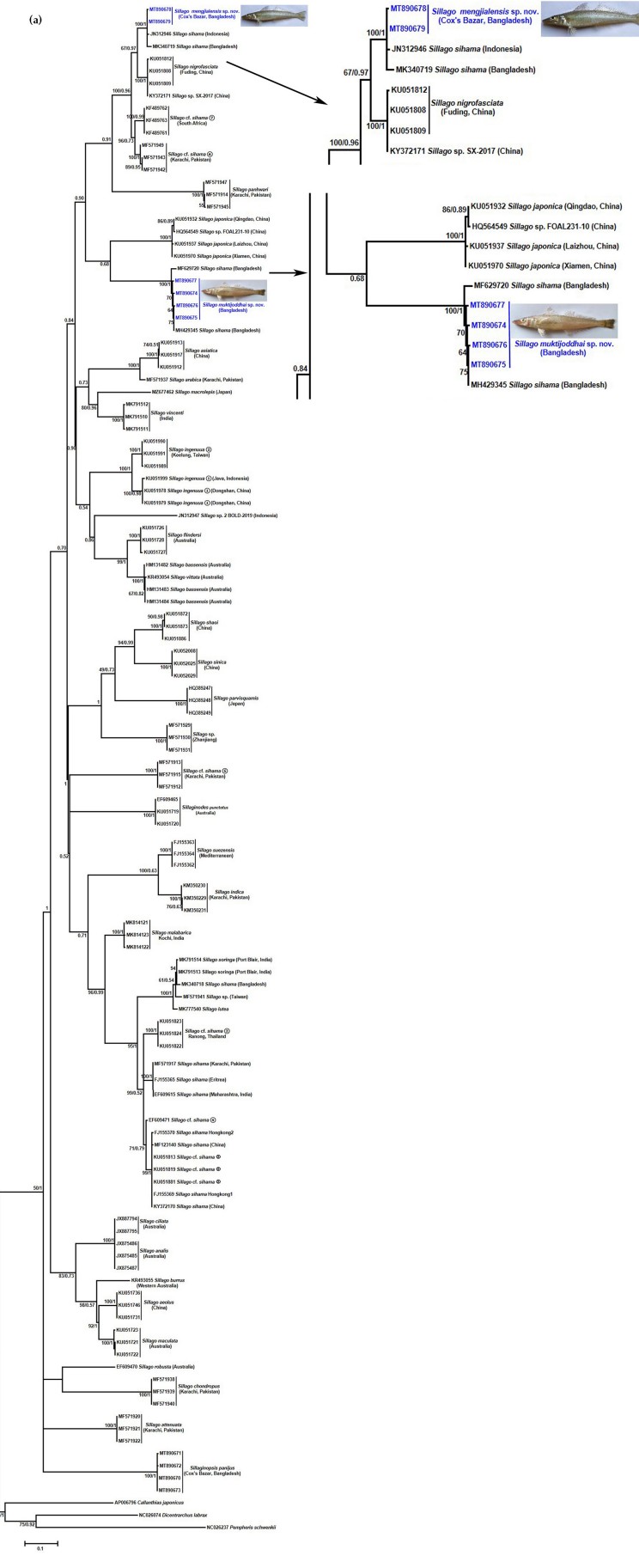

**Figure 4.** *Cont.*

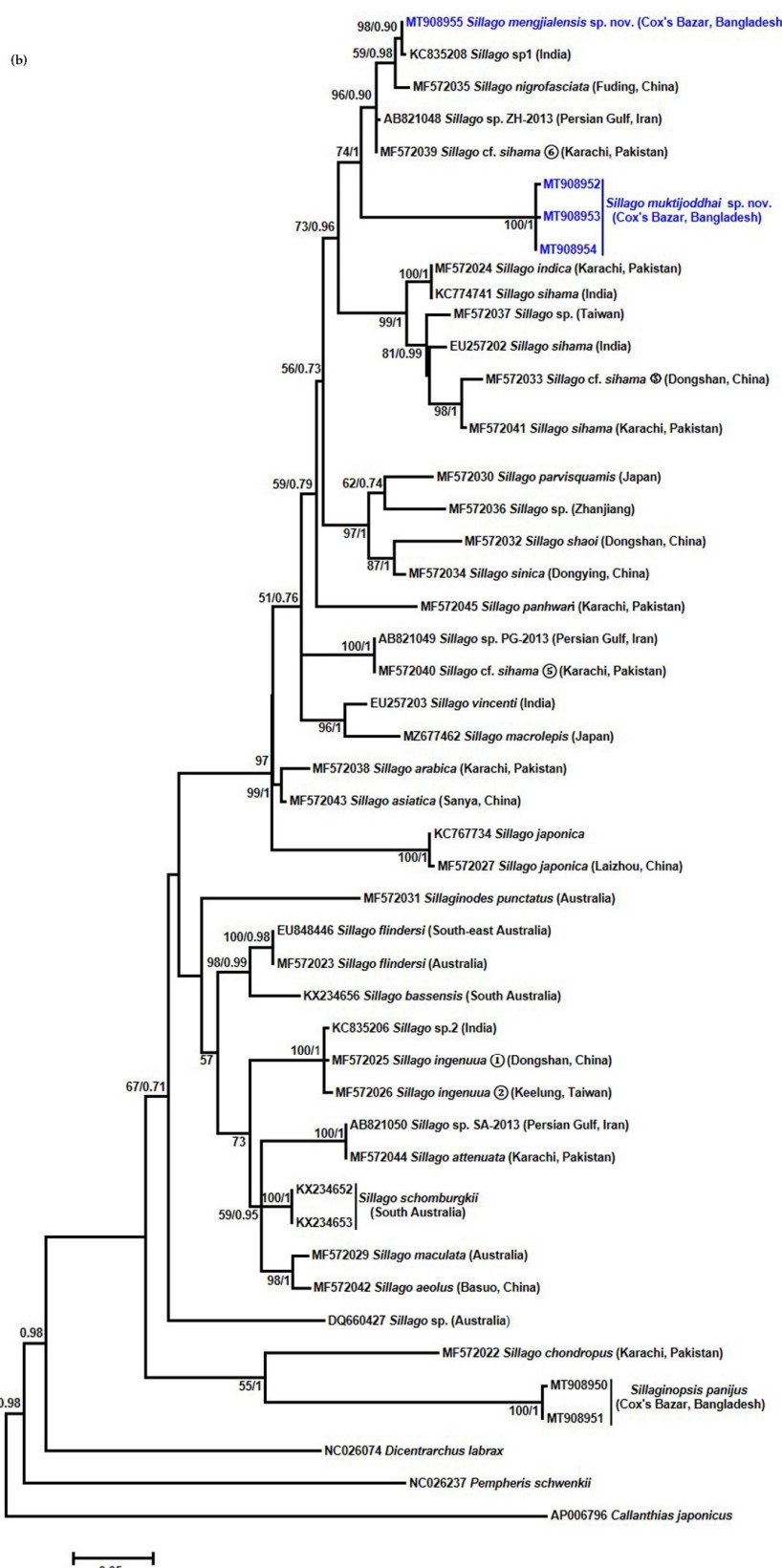

**Figure 4.** Phylogenetic tree using the maximum likelihood analysis and Bayesian phylogenetic analysis method for the species of Sillaginidae. (**a**) COI gene sequences of 42 species; (**b**) 16S rRNA gene sequences of 32 species. The maximum likelihood topology was similar to the result of the Bayesian analysis. Bootstrap support values/Bayesian posterior probabilities are displayed at branch nodes. Three species of the suborder Percoidei, *Callanthias japonicas*, *Dicentrarchus labrax,* and *Pempheris schwenkii*, were selected as outgroup species.

## 4. Discussion

The genetic analysis based on the COI and 16S rRNA sequences of the Sillaginidae species revealed that the genetic divergences ranged from 5.3 to 25.9% and from 1.3 to 22.1% for the two *S*. spp. nov. and other sillaginids, respectively. This result supports the species-level divergence rules proposed by Hebert et al. (2004) and Ward et al. (2009) and demonstrates the effectiveness of using the COI rather than the 16S rRNA gene sequences as a reliable molecular marker by which to identify Sillaginidae species [46,47].

Based on the morphology, the two *S*. spp. nov. differ from the other 14 *Sillago* species (Table 2, Figure 3a–p), especially those having swimbladders with two posterior extensions, and they also showed genetic differences (COI: 5.3–25.9% excluding *S. caudicula*, *S. intermedius* and *S. megacephalus*, due to the unavailability of sequences, 16SrRNA: 1.3–14.7%).

**Table 2.** Comparison of *S. muktijoddhai* sp. nov. and *S. mengjialensis* sp. nov. and 14 other species of *Sillago* with two posterior extensions of the swimbladder.

| Species | Dorsal Fins | Anal Fin | Scales in Lateral Line | Scales above/ below Lateral Line | Gill Rakers First Arch | Vertebrae | Blotches | HL/SL (%) | SLw/HL (%) |
|---|---|---|---|---|---|---|---|---|---|
| *S. muktijoddhai* sp. nov. (*n* = 124) | X–XI,I,20–22 | II,21–23 | 68–72 | 4–6/10–13 | 3–5 + 8–10 = 11–14 | 12–14 + 5–8 + 12–15 = 32–35 | Absent | 21.8–31.3 | 20.5–35.98 |
| *S. mengjialensis* sp. nov. (*n* = 87) | XI,I,20–22 | II,20–23 | 66–72 | 4–5/10–12 | 3–4 + 8–10 = 11–13 | 13–14+5–8+12–15 = 31–35 | Absent | 25–31.7 | 17.3–37.5 |
| *S. indica* [36] (*n* = 72) | X–XI,I,20–22 | II,21–23 | 68–71 | 5–6/10–12 | 3–4 + 7–8 = 10–12 | 33–35 | Absent | 27.5–32.4 | 40.1–46.9 |
| *S. sihama* (Red Sea) [8] (*n* = 11) | XI,I,20–21 | II,21–23 | 70–74 | – | 4+9 | 14 + 2–8 + 12–18 = 34 | Absent | – | – |
| *S.* cf. *sihama* ③ (China) [9] | XI,I,20–23 | II,21–23 | 68–72 | 5–6/10–12 | 3 + 8–9 | 34 | Absent | 24.0–30.0 | – |
| *S. shaoi* [9] (*n* = 39) | XI,I,20–22 | II,21–22 | 70–73 | 5–6/10–12 | 3–4 + 5–6 | 35 | Absent | 26.1–31.0 | 41.8–50.2 |
| *S. suezensis* [8] (*n* = 92) | X–XII,I,19–22 | II,18–22 | 63–74 | – | 3–4 + 8–10 | 13 + 3 + 18 = 34 | Absent | 26.6–27.0 | – |
| *S. sinica* [7] (*n* = 53) | X–XI,I,20–22 | II,21–23 | 75–79 | 5–6/9–11 | 2–4 + 6–8 | 37–39 | Absent | 24.7–29.8 | 33.8–45.1 |
| *S. panhwari* [10] (*n* = 55) | X–XII,I,20–22 | II,18–23 | 69–84 | 3–4/7–10 | 3–4 + 7–9 | 33 | Absent | 27.9–35.0 | 39.5–46.5 |
| *S. caudicula* [6] (*n* = 4) | XI,I,22–23 | II,23–24 | 71 | 5/11 | 4+11 | 14–15 + 6 + 14–15 = 35–36 | Present | 29.0–30.1 | – |
| *S. intermedius* [1] | XI,I,21–22 | II,21–22 | 67–70 | 6–7/8–9 | – | 34 | Present | 30.0–31.0 | – |
| *S. megacephalus* [1] (*n* = 1) | XI,I,22 | II,23 | 70 | 5/10–11 | – | – | Absent | 33.0 | – |
| *S. parvisquamis* [1] | XII–XIII,I,20–22 | II,22–24 | 79–84 | 7/11–12 | 1–2 + 7–9 | 39–40 | Absent | 25.9–27.7 | – |
| *S. nigrofasciata* [11] (*n* = 108) | X–XII,I,20–22 | II,20–22 | 67–75 | 4–6/9–12 | 2–4/5–8 | 34–35 | Black stripe | 25.1–30.8 | – |
| *S. malabarica* [12] (*n* = 34) | XI–XII,I,21–24 | II,22–24 | 68–72 | 4–5/8–9.5 | 3–4 + 6–8 | 13 + 4 + 17 = 34 | Absent | 25–30.4 | 39.4–46.8 |
| *S. parasihama* [13] (*n* = 48) | XI–XII,I,18–21 | II,19–21 | 65–70 | 4–5/8–10 | 2–3 + 5–7 | 14 + 4–7 + 13–16 = 34 | Absent | 18.4–29.0 | 38.9–48.0 |

Notes: References are shown as superscripts next to the species name, n denotes the number of individuals examined and ③ means cryptic species in the *Sillago sihama* complex.

*S. muktijoddhai* sp. nov. is similar to *S. indica* based on the countable characters and swimbladder structure, but striking differences exist, such as fewer dark spots on the skin and fins and a larger number of gill rakers. In addition, there is a short snout and a dissimilar swimbladder. The posterior sub-extension of the anterolateral extensions of the swimbladder is strong for about half of its length towards the beginning, and the remainder is simple and thin in *S. muktijoddhai* sp. nov., but thin for the whole length of the extension in *S. indica* (Figure 3c,e). There is also a large genetic distance (COI, 22.7% and 16SrRNA, 10.5%) between them.

*S. mengjialensis* sp. nov. was previously misidentified as *S. sihama*, but striking differences exist, such as black dots on the anal fin, and a dissimilar swimbladder. The posterior sub-extension of the anterolateral extensions of the swimbladder is thin in *S. mengjialensis* sp. nov. and thick in *S.* cf. *sihama* ③ China (Figure 3d,f), and a large genetic

distance (COI, 17.2% and 16SrRNA, 8.1%) exists in between. Moreover, *S. sihama* (Southern Red Sea) lacks an anterior sub-extension of the anterolateral extension, and two posterior tapering extensions are separated from each other (Figure 3g) but are different from those in *S. mengjialensis* sp. nov. These differences include the presence of an anterior sub-extension of the anterolateral extension and two posterior extensions without separation from each other. However, a large genetic distance (COI, 18.8% and 16SrRNA, 6.9%) exists in between.

*S. muktijoddhai* sp. nov. and *S. mengjialensis* sp. nov. are similar in appearance, but their differentiation should be based on body color, swimbladder structure (Figure 2a,b and Figure 3a–d), and genetic distance (COI: 18.2%, 16SrRNA: 8.2%).

The *S. sihama* complex speciated after the rise of sea levels following the last ice age, which expanded shallow inshore habitats with isolating mechanisms resulting in populations of this complex. We propose that additional species will be discovered. In the present study, large genetic distances and extensive morphological studies can differentiate *S. muktijoddhai* sp. nov. and *S. mengjialensis* sp. nov. from their related species. Their description from the Northern Bay of Bengal, the coast of Bangladesh, highlights the need for more studies on their ecology, distribution, and abundance patterns, which are essential for the efficient resource management and conservation of ecologically and economically important sillaginids in the Bay of Bengal on the coast of Bangladesh.

Key to *Sillago* species (swimbladder with two posterior extensions), extended and modified after Xiao et al. (2016) [9]

1. Body with dark blotches or spots ................................................................................................ 2
   - Body without dark blotches or spots ......................................................................................... 3
2. Anal fin rays ≥ 23; vertebrae more than 34 ...................................................... *S. caudicula*
   - Anal fin rays less than 23; vertebrae 34 .................................................... *S. intermedius*
3. HL/SL less than 33% ..................................................................................................................... 4
   - HL/SL is 33% .............................................................................................. *S. megacephalus*
4. Second dorsal fin membrane with many rows of clear dusky spots ............................... 5
   - Second dorsal fin membrane without any row of dusky spots ............................... 6
5. Second dorsal fin with 5 or 6 rows of dusky spots along with rays ...... *S. parvisquamis*
   - Second dorsal fin with 3 or 4 rows of dusky spots along with rays ......... *S. sinica*
6. Anal fin membrane hyaline ......................................................................................................... 7
   - Anal fin membrane usually with spots ..................................................................................... 8
7. Preopercle and most of the opercle without scales ........................................ *S. suezensis*
   - Preopercle and most of the opercle with scales ............................................................ 9
8. Gill rakers on the first arch 3–4/7–8; posterior sub extension of anterolateral extensions of swim bladder thin; roots of two posterior extensions not adjoining and two posterior extensions not adjoining ....................................................................................... *S. indica*
   - Gill rakers on the first arch 3–4/5–6; posterior sub extension of anterolateral extensions of swim bladder strong; roots of two posterior extensions not adjoining and two posterior extensions not adjoining .................................................. *S. shaoi*
   - Gill rakers on the first arch 3–5/8–10; posterior sub extension of anterolateral extensions of swim bladder strong for about a half-length towards beginning, then the remainder is simple and thin; roots of two posterior extensions not adjoining and two posterior extensions not adjoining ........ *S. muktijoddhai* sp. nov.
   - Gill rakers on the first arch 2–4/5–8; posterior sub extension of anterolateral extensions of swim bladder strong; roots of two posterior extensions adjacent and two posterior extensions in close proximity ............................ *S. nigrofasciata*
   - Gill rakers on the first arch 3–4/8–10; posterior sub extension of anterolateral extensions of swim bladder thin; roots of two posterior extensions adjoining and two posterior extensions in close proximity ...................... *S. mengjialensis* sp. nov.

9.    Anterior extensions not joined at the origin; lacks an anterior sub-extension of the anterolateral extension; no lateral processes; two posterior tapering extensions are separated from each other ................................................... *S. sihama* (Southern Red Sea)

-    Snout length 40.4–45.5% of head length; anterior extensions not joined at the origin; nine to ten lateral processes; the base of two posterior extensions adjoining and two posterior extensions adjoining ............................. *S.* cf. *sihama* ③ (China)

-    Snout length 39.4–46.8% of head length; anterior extensions not joined at the origin; eight to nine lateral processes; the base of two posterior extensions not adjoining and two posterior extensions not adjoining ........................ *S. malabarica*

-    Snout length 39.5–46.5% of head length; anterior extensions joined at the origin; no lateral processes; two narrow posterior extensions separated from each other ............................................................................................................ *S. panhwari*

-    Snout length 38.9–48.0% of head length; posterior sub extension of anterolateral extensions with some dwarf blind tubules, one-sided and outward and about one-third to half-length of the swim bladder; two dumpy posterior extensions separated from each other ................................................................. *S. parasihama*

**Supplementary Materials:** The following supporting information can be downloaded at: https://www.mdpi.com/article/10.3390/fishes7030093/s1, Figure S1: X-ray photographs of two new *Sillago* species from the Bay of Bengal, Bangladesh. (**a**) *Sillago muktijoddhai* sp. nov., CO12059, paratype; (**b**) *Sillago mengjialensis* sp. nov., CO12068, paratype; Figure S2: Phylogenetic tree using maximum likelihood analysis and Bayesian phylogenetic analysis method for the species of Sillaginidae. (**a**) COI gene sequences of 42 species; (**b**) 16S rRNA gene sequences of 32 species. The maximum likelihood topology was similar to the result of the Bayesian analysis. Bootstrap support values/Bayesian posterior probabilities are displayed at branch nodes. Three species of the suborder Percoidei, *Callanthias japonicas*, *Dicentrarchus labrax,* and *Pempheris schwenkii*, were selected as outgroup species. Table S1: Sampling information of two new *Sillago* species from the Bay of Bengal, Bangladesh; Table S2: Details of specimens used in genetic analysis with individual ID, sampling location, gene information, GenBank accession numbers, and references. Table S3. Net genetic distances (K2P) for COI gene sequences of the 42 *Sillago* species. Table S4. Net genetic distances (K2P) for 16S rRNA gene sequences of the 32 *Sillago* species.

**Author Contributions:** Conceptualization, T.G.; methodology, S.S. and M.A.B.; software, S.S. and Z.Y.; validation, T.G. and S.S.; formal analysis, S.S.; investigation, N.S. and M.A.B.; resources, T.G. and N.S.; data curation, Z.Y. and S.S.; writing—original draft preparation, T.G. and S.S.; writing—review and editing, R.J.M. and J.Q.; visualization, T.G.; supervision, T.G.; project administration, T.G.; funding acquisition, T.G. All authors have read and agreed to the published version of the manuscript.

**Funding:** This study is supported by the National Natural Science Foundation of China (No. 41976083).

**Institutional Review Board Statement:** Not applicable.

**Data Availability Statement:** The COI and 16S rRNA gene sequences generated in this study have been submitted to the GenBank. These data can be found here: [https://www.ncbi.nlm.nih.gov/nucleotide/], accessed on 2 November 2021.

**Acknowledgments:** We are grateful to Kishor Kumar Sarker and Abdullah Al Mamun for the collection of some samples. We are grateful to Palash Ahmed, Md. Anwar Parves, and Prianka Kundu for their help in taking morphological measurements and Irin Sultana and Jia-Guang Xiao for some technical support.

**Conflicts of Interest:** All the authors declare that they have no conflict of interest.

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
