# Peer review of "Descriptions of Two New Species, Sillago muktijoddhai sp. nov. and Sillago mengjialensis sp. nov. (Perciformes: Sillaginidae) from the Bay of Bengal, Bangladesh"

_fishes, doi:10.3390/fishes7030093_

Round 1
Reviewer 1 Report
This manuscript used morphological characters and genetic analyses to justify the identification of two new species, Sillago muktijoddhai sp. nov. and Sillago mengjialensis sp. nov. (Perciformes: Sillaginidae), from the Bay of Bengal, Bangladesh. The paper is interesting and potentially useful, as it increased the number of recognized species of Sillago in the world, and made an attempt to confirm the prevailing misidentification of the two new species in Bangladesh as so-called S. sihama. I think that the integrated approach including morphology and DNA barcoding in a maximum likelihood analysis and Bayesian phylogenetic analysis framework is an interesting advantage of this paper in relation to others that deal with similar topics. There are some justifications the authors need to include, which will enrich the content of the research while clarifying the selection and implementation of the approaches used. The specific comments are:
Abstract
- (1) Page 1 Line 15. “Bangladesh integrated” to “Bangladesh, integrated”.
- (2) Page 1 Line 21 (and others throughout the manuscript). I would suggest using “14” rather than “fourteen” when the number is more than ten.
- (3) Page 1 Line 23. “support” to “supported”.
- (4) Page 1 Line 24. “new species” to “the new species”.
Introduction
- (5) Page 1 Line 34. “S. sihama and” to “S. sihama, ”.
- (6) Page 1 Line 36. “one species” to “and one species”.
- (7) Page 1 Line 36. “S. panijus are” to “S. panijus, are”.
- (8) Page 1 Line 41. “and molecular” to “or molecular”.
- (9) Page 2 Line 50. “extensions arises” to “extensions arise”.
- (10) Some additional information such as brief biotic background (e.g., predators, prey, and competitors) on the family Sillaginidae or the five genera in the study area would be helpful in the Introduction section to set the stage. I would also suggest emphasizing somewhere in the Introduction section the significance of integrated approach including morphology and DNA barcoding in a maximum likelihood analysis and Bayesian phylogenetic analysis framework, because these approaches might help people from broader field find your work useful rather than just people working on the particular species or area being interested in it.
Materials and Methods
- (11) Page 2 Lines 75-79. Please check the grammar for this over-long sentence. I would suggest breaking it into two shorter sentences.
- (12) Page 3 Line 92. “twenty-six” to “26”.
- (13) Page 3 Line 97. “while” to “, while”.
- (14) Page 3 Lines 112-113. There is little explanation as to why maximum likelihood analysis or Bayesian phylogenetic analysis was chosen in this study. There have been some other modeling approaches that were used in evolutionary biology when it comes to analyzing phylogenetic relationships. However, there is little justification as to why the authors only chose these methods. I would suggest adding a brief explanation to justify why these approaches and not other also commonly used ones were chosen. Given the potential of modeling methods to influence the validity of the new species, it would be better to critically assess why some approaches are chosen and their potential and weaknesses for the available data and specific environment considered.
- (15) Page 3 Lines 113-117. There is little explanation for how Bayesian phylogenetic analysis was performed in this study. I would suggest providing a flowchart and a table describing the Bayesian model, parameters, and how robust values for priors in the Bayesian model were selected. A Bayesian flowchart typically uses squares, ellipses and arrows to show how variables and parameters are related through conditional probabilities. A table could be used to provide the descriptions of parameters in the Bayesian model (e.g., observed/unobserved fixed variables, observed/unobserved random variables, random factors, and hyperparameters). How robust values for priors in the Bayesian model were selected is particularly important, as priors can contribute the information needed to fill the gaps in scenarios where there are sparse or missing data.
Results
- (16) Page 4 Line 127. “SL” to “standard length (SL)”. You may want to specify the full terms for abbreviations the first time they appear in the text, because people who are not in this field may not be familiar with them.
- (17) Besides abiotic conditions of the species’ habitats, I would suggest adding some brief information on their biotic habitats (e.g., potential predators, prey, and competitors) in the Results or Discussion section based on observation/literature and their potential influence on the distributions of the two new species in the study area.
Discussion
- (18) Page 12 Line 313. “for two” to “for the two”.
- (19) Page 12 Line 318. “fourteen” to “14”.
- (20) Page 12 Line 319. “extensions” to “extensions, ”.
- (21) Page 12 Line 320. “and also genetic differences” to “and they also showed genetic differences”.
- (22) Page 13 Line 325. “The S. muktijoddhai” to “S. muktijoddhai”.
- (23) Page 13 Line 327. “more number of” to “larger number of”.
- (24) Page 13 Line 333. “The S. mengjialensis” to “S. mengjialensis”.
- (25) Page 13 Line 337. “the S. sihama” to “S. sihama”.
- (26) Page 13 Line 347. “The S. sihama” to “S. sihama”.
- (27) Page 13 Lines 347-349. Please check the grammar for this over-long sentence. I would suggest breaking it into two shorter sentences.
- (28) Page 13 Line 349. “and we propose” to “, and we propose”.
- (29) Page 13 Line 353. “more study” to “more studies”.
Tables and Figures
- (30) Page 3 Figure 1 Line 90. “are provided” to “is provided”.
- (31) Page 5 Figure 2 Line 152. “157.84 mm SL” to “157.84 mm standard length (SL)”. I would suggest using “standard length (SL)” rather than “SL” in the figure caption, as audience who are not in this field may not be quite familiar with the abbreviation. A good figure or table caption should make the figure or table understandable without reference to the main text.
- (32) Page 6 Table 1 Line 172. “Bay of Bengal” to “the Bay of Bengal”.
- (33) Page 8 Figure 3 Line 228. “[12],” to “[12]; ”.
- (34) Page 12 Figure 4 Line 306. “ML and BI method” to “maximum likelihood analysis and Bayesian phylogenetic analysis”.
- (35) Page 12 Table 2 Line 322. “fourteen” to “14”.
Reviewer 2 Report
The authors of this manuscript describe the problem of identifying species of the genus Sillago and present a way to solve it based on the methods of classical and molecular analysis. The work provides tools that also allow other researchers to correctly identify these fish species.
The language of the manuscript is clear, proper research methods have been used, and the references are appropriate. However, some parts of the manuscript require changes (giving below):
- (Introduction, line 34): "(...) 39 species and five genera [3-13]" - Can the authors reduce the number of cited sources to a few of the most important (if possible)?
- In the last sentence of the introduction (lines 66-68), the authors refer to fishery management, biodiversity maintenance and resource exploitation regarding the described species. However, we do not find any information about the ecological or economic status of these species beforehand. I think it is advisable for the readers to know why the identification of the mentioned species is important.
- (MM, Figure 1): The figure is not properly embedded in the text (broken sentence - lines 86-87).
- (MM, Figure 1, lines 88-90): Figure caption requires correction - no reference to the abbreviation "SP1".
- (MM, lines 100-101): The description of the reaction mixture is imprecise - the authors provide the volumes without specifying the concentrations of individual components (dNTP, Taq polymerase, DNA template, primers).
- (MM, lines 109-110): "(...) was used for sequence 109 comparison". It seems to me that the authors mean more forward and reverse sequence alignment.
- (MM, lines 118-120): "In the present study, eleven haplotype COI and seven haplotype 16S rRNA sequences were obtained from two new Sillago species, Sillaginopsis panijus and Sillago macrolepis.". This sentence is unclear - were the mentioned haplotypes unique to the listed species? Moreover, are these the results? If so, why have they been included in the 'Materials and methods' section?
- (Results, Figure 4): I wonder about the point of including this plot in the main text of the manuscript. The size of the trees makes them practically unreadable and therefore of little help in direct data analysis. I propose to put this figure in the Supplementary Materials section.
- (Discussion, lines 356-403): I also suggest moving the 'Key to Sillago species' to the Supplementary Materials section.
Round 2
Reviewer 1 Report
The authors have addressed most of the comments. I just have a few minor suggestions below, which I hope could help further improve the clarity of the paper.
For the revised-highlighted version:
- (1) Page 1 Line 36. “and S. chondropus” to “S. chondropus”. Please remove the “and”.
- (2) Page 3 Line 115. “As Maximum” to “Maximum”. I would suggest removing the “As”.
- (3) Page 3 Line 116. “information and least affected” to “information, and is least affected”.
- (4) Page 3 Line 117. “error, and posterior probability” to “error. The posterior probability”.
- (5) Page 3 Line 119. “Maximum” to “maximum”.
- (6) Page 13 Line 318. “Maximum” to “maximum”.
- (7) Page 13 Line 319. I would suggest removing the “method”.
- (8) Page 14 Line 360. “ice age. Which” to “ice age, which”.
- (9) Page 15 Line 420. “Maximum” to “maximum”.
- (10) Page 15 Line 421. I would suggest removing the “method”.
